# Development and Implementation of Multidisciplinary Liver Tumor Boards in the Veterans Affairs Health Care System: A 10-Year Experience

**DOI:** 10.3390/cancers13194849

**Published:** 2021-09-28

**Authors:** Atoosa Rabiee, Tamar Taddei, Ayse Aytaman, Shari S. Rogal, David E. Kaplan, Timothy R. Morgan

**Affiliations:** 1Washington DC VA Medical Center, 50 Irving Street, NW, Washington, DC 20422, USA; 2VA Connecticut-Healthcare System, 950 Campbell Ave, West Haven, CT 06516, USA; tamar.taddei@va.gov; 3VA New York Harbor Health Care System, 423 E 23rd St, New York, NY 10010, USA; ayse.aytaman@va.gov; 4VA Pittsburgh Healthcare System, 4100 Allequippa St, Pittsburgh, PA 15240, USA; shari.rogal@va.gov; 5Corporal Michael J. Crescenz VA Medical Center, 3900 Woodland Ave, Philadelphia, PA 19104, USA; David.kaplan2@va.gov; 6Division of Gastroenterology and Hepatology, Hospital of the University of Pennsylvania, 3400 Sprice St, Philadelphia, PA 19104, USA; 7VA Long Beach Healthcare System, 5901 E 7th St, Long Beach, CA 90822, USA; timothy.morgan@va.gov

**Keywords:** hepatocellular carcinoma, tumor board, veterans affairs

## Abstract

**Simple Summary:**

In this perspective piece we will summarize our path to implement local, regional and integrated service network tumor boards and our use of technology to facilitate this process and provide guidance for other institutions outside of Veterans Affairs system to implement this unique model. We offer a road map to establish regional multidisciplinary liver tumor boards.

**Abstract:**

In this perspective piece, we summarize the development and implementation of multidisciplinary liver tumor boards across the Veterans Affairs health care system dating back to 2010. Referral to multidisciplinary tumor boards (MDLTB) has been demonstrated to decrease the number of unnecessary invasive procedures, reduce health care costs and maximize patient outcomes. Although the VA is the largest single care provider in the US, there is significant heterogeneity in healthcare delivery. We have shown that receiving care at VA centers with MDLTB is associated with higher odds of receiving active therapy and a 13% reduction in mortality. Access to expert hepatology care appears to be one of the critical benefits of MDLTB resulting in 30% reduction in mortality. Integrated health care systems such as the VA have the unique capability of implementing virtual tumor boards that can easily overcome geographic barriers and standardize care across multiple facilities regardless of their access to hepatology or other disciplines. Significant barriers remain requiring implementation plans. This document serves as a roadmap to establish multidisciplinary tumor boards, including standardization of imaging reports, identifying stake holders who need to be present at tumor board, institution buy-in, and specifics for local, regional and integrated service network tumor boards.

## 1. Background

Liver cancer is the sixth most commonly diagnosed cancer and the fourth leading cause of cancer death [1]. Hepatocellular carcinoma (HCC) accounts for 85% of primary liver cancers with the majority of the remainder resulting from bile duct (cholangiocarcinoma) or mixed tumors. While HCC incidence was projected to rise through 2030 [2], recent data show the rates are starting to decline [3], likely predominantly explained by widespread treatment of chronic viral hepatitis.

Care by a multidisciplinary team and discussion of cases at a multidisciplinary tumor board have been shown to improve outcomes for patients with liver cancer. In a retrospective study of nearly 4000 patients with HCC in the Veterans Administration (VA) healthcare system, a multidisciplinary approach to delivering care was associated with improved overall survival [4]. In another study comparing patients with HCC managed by a multidisciplinary tumor board (MDTB) to those who were not, it was demonstrated that patients managed by MDTB were more likely to present with cancer at earlier stages and with lower serum alpha-fetoprotein (AFP). These patients also had higher odds of receiving any treatment for HCC, independent of their model for end-stage liver disease (MELD) score, serum AFP, and tumor stage. This study also showed a significant survival benefit for patients managed through MDTB [5]. Another study showed that discussion of patients at MDTB doubled referral rates for treatment, improved early-stage detection, increased receipt of curative therapy and palliative care, and improved overall survival [6,7,8]. Furthermore, data support the effect of MDTB on improving the patient experience [9].

Hepatocellular carcinoma almost always arises in the setting of cirrhosis, which can vary widely in severity. A multidisciplinary approach is imperative to guide the management plan with consideration of severity of underlying liver disease, performance status, tumor stage, and presence of metastatic disease.

Multiple providers such as hepatologists and gastroenterologists, interventional and diagnostic radiologists, surgical oncologists and transplant surgeons, medical oncologists, radiation oncologists, pathologists, and primary care and palliative care providers should be involved in the care of patients with HCC. Ancillary services, such as nutrition, social work, mental health, and patient support groups also play a vital role in patient care [10].

Here, we discuss our experience with development and implementation of multi-site, virtual MDTBs that provide recommendations for patients with HCC in geographically diverse medical facilities within the VA healthcare system.

## 2. History of Liver Cancer Care at the VA

The Veterans Health Administration (VHA) oversees the largest integrated healthcare system in the United States, including 170 VA medical centers (VAMCs) and 1074 outpatient sites caring for varying complexity of diseases and serving 6 million enrolled Veterans each year. Since VAMCs cross urban, suburban, and rural settings and include varying levels of specialty care, access to subspecialists, ancillary services, and multidisciplinary tumor boards may vary geographically.

The VA is separated geographically into 23 regional Veterans Integrated Service Networks (VISN), each of which spans several states. VISN leadership evaluates and provides resources for each individual facility in their jurisdiction. This gives each VISN the unique opportunity to budget and plan Veterans’ healthcare for a particular geographic area. Increasing access to care in the VA is a national goal and VA has been proactive in the use of technology to overcome geographic barriers to standardize care across multiple facilities regardless of academic affiliation and/or distance from tertiary care centers.

Determinants of care for HCC in the VA have been assessed by a large study which can serve as a benchmark by which to set metrics for healthcare delivery. The Veterans Outcomes and Costs Associated with Liver disease (VOCAL) Study Group developed an HCC cohort including 3988 patients diagnosed with liver cancer between 2008–2010 identified by ICD9-CM diagnosis codes and verified by chart review. Most patients were treated in an urban setting and half lived within 74 miles of the closest VA hospital. Among these patients, 69% received care at academically affiliated VA medical centers and 51% were treated at a VA medical center more than 500 miles to the nearest VA transplant center. Only 34% were discussed at a multidisciplinary tumor board conference. This number further decreased in patients with Child Pugh B or C cirrhosis.

In this heterogenous group of patients, only 74% of patients received active HCC treatment. Furthermore, among patients with potentially curable HCC (Barcelona-Clinic Liver Cancer BCLC stages 0/A), only one out of four patients received potentially curative therapy. In this study, facility rurality and patient distance to VA medical centers had no significant impact on receiving HCC therapy; however, patients treated in VA centers with an academic affiliation had higher odds of receiving active HCC therapy. Patients evaluated by gastroenterology and palliative care, rather than hepatology or surgery, had lower odds of receiving active HCC therapy. Overall survival in this cohort was associated with seeing a specialist within 30 days of diagnosis and management by multidisciplinary tumor board [4]. Similar observations have been made in non-VA medical centers with improvement of patient survival after establishment of MDTB [8,11].

In lockstep with this research, and with the support of the VA HIV Hepatitis and Related Conditions Program (HHRC), clinical champions across the VA, including members of this study group, worked collaboratively to operationalize regional networks for HCC care and to consider how best to set standards and metrics of quality care (Figure 1).

## 3. Components of Multidisciplinary Tumor Board

Diagnosis and management of HCC is highly specialized and available only in select facilities in the VA system. This disparity in access to specialty care led to the creation of regional tumor boards to expand availability of highly specialized expertise and treatment.

Several steps were followed in implementing regional tumor boards throughout VHA. First the needs were established related to geographical gaps in specialty care. Initiation of a multidisciplinary and/or multifacility tumor board started with gaining leadership support, presenting data to support gaps where outcomes could improve, and developing a business case to justify the necessary resources. According to National Comprehensive Cancer Network (NCCN) guidelines [12], liver cancer tumor boards must be attended by key disciplines, including oncology, radiation oncology, GI/hepatology, surgery, diagnostic radiologist, interventional radiology, and palliative care. However, the broader team involves cancer registrars, cancer navigators, social workers, and psychologists and extends to support staff, such as clinical application coordinators, telehealth coordinators, and program assistants. Creation of a shared electronic space, e-mail groups, and dedicated smaller work groups can aid greatly in the standardization and quality improvement of multiple key processes not only for single institution tumor boards but also for multi-facility boards.

Liver cancer is the only solid organ malignancy that can be diagnosed exclusively by imaging. Therefore, proper image acquisition and quality are essential elements of liver cancer management. Standardization of liver imaging protocols across the VISN for contrast enhanced CT and MRI as well as sonograms improved accuracy and reporting. Radiology templates were upgraded with eventual inclusion of the liver imaging reporting and data system (LIRADS) [13].

Building a team composed of faculty from multiple institutions requires considerable diplomacy and skill. Teams are psychodynamic systems and need a common goal, trust, a sense of inclusion, delineation of boundaries, and clear guidance for conflict resolution. The key to success is for every single team member to have a sense of ownership and accountability. Annual VA Liver Cancer Summits were initially regional, but eventually national, and every team member’s involvement not only improved knowledge and enhanced quality of care, but also led to ongoing maturation of teams that began to serve as examples for other centers and regions. Team creation workshops were added to annual meetings to reinforce the skills of high functioning teams and to help budding teams across the nation.

A checklist for development of MDLTB is included as Table 1.

The first VISN-wide liver cancer tumor board was created in 2012 with HHRC funding and support. This multifacility tumor board (in VISN 2) includes five centers affiliated with major Universities and one spoke facility without University affiliation. Liver cancer expertise was available in three of the five facilities, each offering select treatments. Supported by VISN and facility leadership, program infrastructure was developed to facilitate communication (e-mail groups with key stakeholders, shared network space), referral (interfacility consultations, both virtual and in person), and documentation (standardized templates for documentation informed by NCCN guidelines).

In 2018, building on national efforts at quality improvement, a VA collaborative consensus statement on a pathway for the development of a multidisciplinary team to manage HCC) was written to set forth best practices in the VA regarding diagnosis and staging of HCC and MDTB composition and requirements, defining key elements of tumor board documentation, surveillance intervals, and surveillance duration after curative and palliative treatment [14].

In this document and based on the American College of Surgeons (ACOS) guidelines, a 75% adherence rate with participation at tumor board among the required disciplines was advised. The disciplines required to participate in MDLTB are shown in Figure 2. The operations of MDLTB should be reviewed annually by the institutional cancer committee. Elements to consider in formation of MDLTB include: frequency and format of the conference, disciplines attending, case content such as newly diagnosed, need for additional treatment or adjuvant treatment, disease recurrence or progression, and discussion of palliative care. A patient information tool to help navigate the complex treatment options of HCC has also been developed [15]. 

The described efforts resulted in the establishment of tumor boards across VHA (Figure 3). These are at different stages of development serving Veterans locally and regionally.

The VISN 2 team demonstrated significant improvement in timeliness of care and survival. Median time to initial HCC treatment at all sites was 39 days and improved to 26 days after implementation of the MDTB (*p* = 0.052). One-year survival improved from 44% to 67%, and 2-year survival improved from 31% to 49% after implementation of MDTB. As shown in Figure 4, prior to establishment of MDTB, median survival was 9.4 months (95%CI 4.2–14.6), which improved to 23.4 months (95%CI 19.7–27.1) after establishment of MDTB. Although the study was limited due to small sample size, it showed that timeliness of care was significantly improved among all facilities [16]. There was no significant difference in 5-year survival rate in the two groups.

## 4. Discussion and Future Directions

Veterans Affairs hospitals are a unique system that can share medical information despite geographical distance and overcome physical barriers with the use of technology to improve access to care.

We have used many of these unique strengths to our advantage to help with implementation of local and regional tumor boards.

One of the main factors that has led to successful implementation of this program has been the engagement of clinical champions locally and meeting each center’s needs at their current state. As can be seen in this map (Figure 3), there are tumor boards across the country at different stages of development serving veterans locally or at a regional level.

Despite the progress that has been made, there are multiple areas that need improvement in the future. Developing value-based care metrics to help track outcomes and monitor progress as well as to identify barriers will be key in resource management.

Cancer care navigators, especially when care is not being managed at the local level, are paramount to helpful transition of care and timely management of HCC.

Outside of VA, technology integration to help electronic medical records communicate with one another and exploring reimbursement for time spent discussing care of patients with HCC and offering expertise in the setting of a multidisciplinary tumor board would remain challenging.

In summary, VHA leveraged a national data and learning collaborative infrastructure to establish a network of highly successful local and regional tumor boards. The resources developed to support this work can potentially inform implementation in other complex healthcare systems to improve the care of patients with chronic liver disease.

Ultimately, we hope that our experience with establishment of local and regional tumor boards at the VA and use of technology to overcome physical barriers would be a helpful guide to others caring for complex patients with liver cancer to implement such systems at their own health care facilities.

## Figures and Tables

**Figure 1 cancers-13-04849-f001:**
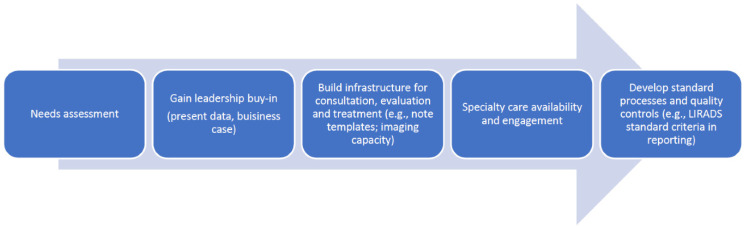
Steps to develop a multidisciplinary tumor board.

**Figure 2 cancers-13-04849-f002:**
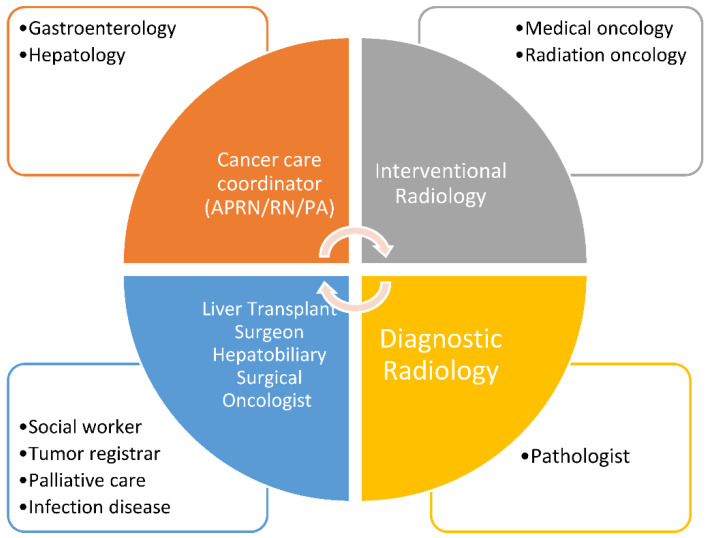
Proposed disciplines to create MDTB to care for liver cancer patients.

**Figure 3 cancers-13-04849-f003:**
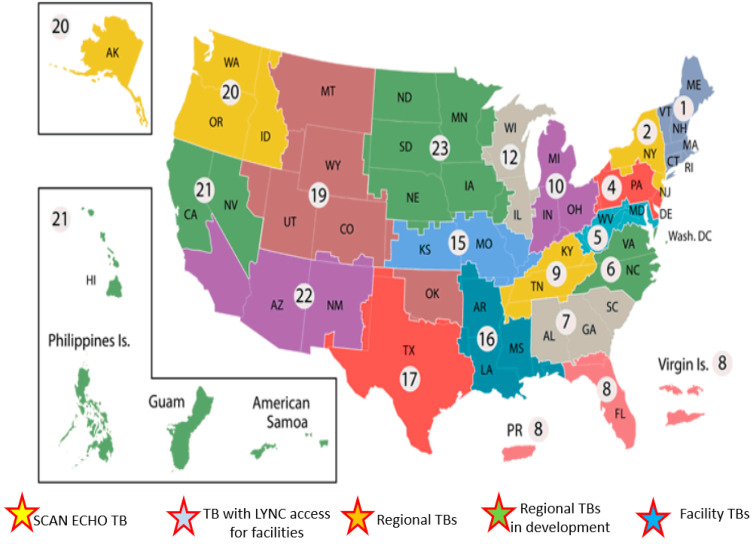
Current state of regional and local liver cance tumor boards in Veterans hospital across the US.

**Figure 4 cancers-13-04849-f004:**
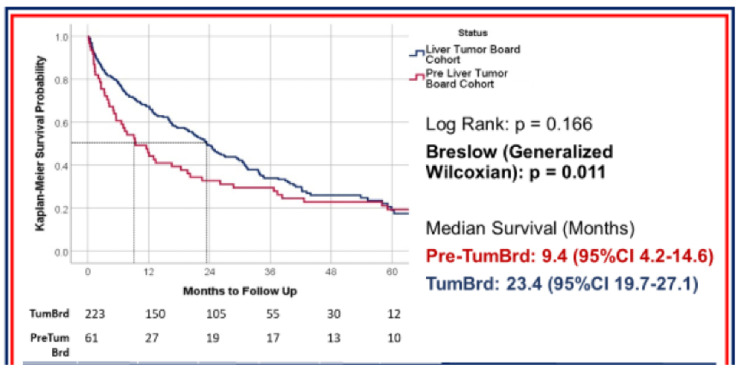
Median survival of patients with liver cancer in VISN 2 before and after implementation of MDTB.

**Table 1 cancers-13-04849-t001:** Checklist of steps for development of MDLTB.

Elements of MDLTB	Key Questions and Possible Solutions
Governance, Recording and Documentation	Where will MDLTB meet? When will MDLTB meet? Who will present cases? Who will record findings, discussion, and recommendations?
Clinical documentation and coordination	MDLTB Common Data Elements Getting data into clinical record – Interfacility consult – Encrypted email to submitter or site coordinator Who is going to communicate with the patient?
Agenda creation and submission	Who will create agenda? Will pre-review be required, who will do? How will incomplete submissions be handled? – Service agreements How/when will agenda be distributed? – How much time do participants need ahead of meeting? – Plan for ad hoc cases
	Getting data into clinical record – SCAN-ECHO or IFC Consult – Encrypted email to submitter or site coordinator – Who is going to communicate with the patient? – Telehealth component?
Case submission	Who will have access to submit? – Open – Limited group based on participation Who will control access? How will data be submitted? – Paper form – Encrypted email – Interfacility Consult – Sharepoint – Service agreements What data are required? See suggested data from VA consensus form Cutoff date/time
Follow up	Computerized Patient Record System (CPRS) orders at Hub – who is going to be responsible? – IR orders – Consults – Admissions Travel vs. Community Care – Does infrastructure exist for facility to facilitate transportation? – Who is responsible for arranging? – Who will track arrangements centrally? Follow-up imaging – Communicating recommendations – Responsibility for completion and resubmission
Audit and feedback	What outcomes are important for MDLTB to track? – Type of treatment – Time to treatment – Complications – Survival Regular review of outcomes How sentinel events will be reported/discussed Reporting requirements (if any)

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
