# Peer review of "Development and Implementation of Multidisciplinary Liver Tumor Boards in the Veterans Affairs Health Care System: A 10-Year Experience"

_cancers, 2021, doi:10.3390/cancers13194849_

Round 1

Reviewer 1 Report

Thank you for letting me review the manuscript (3rd time!!!).

The authors applied some minor changes mentioned by the reviewer team. However, the lack of care is disturbing. Just a few examples:

Figure 3 has typos: "interventioanl" and "patologist". 

Figure 2 is now correctly linked in the main text, however it is the last figure of the manuscript and should be therefore numbered differently.

I cannot see any point-by-point response to the last review round and reviewer comments were implemented only half-heartedly.

Author Response

point by point responses

Figure 3 has typos: "interventioanl" and "patologist". 

The typos have been corrected.

Figure 2 is now correctly linked in the main text, however it is the last figure of the manuscript and should be therefore numbered differently.

We have now submitted each figure as a separate document to help with downloading them in order. 

I cannot see any point-by-point response to the last review round and reviewer comments were implemented only half-heartedly.

please see above for point by point response. We apologize if you did not receive our previous point by point comments, it was uploaded to the journal website. 

Reviewer 2 Report

The authors did not address all previous comments. For example, the authors need to provide references for the first VISN wide liver cancer tumor board -- multifacility tumor board (in VISN 2).

Author Response

point by point response

The authors did not address all previous comments. For example, the authors need to provide references for the first VISN wide liver cancer tumor board -- multifacility tumor board (in VISN 2).

This study has not been published as a manuscript, it has been presented as an abstract in AASLD, the reference for which has been provided. We hope this answers your question.

This manuscript is a resubmission of an earlier submission. The following is a list of the peer review reports and author responses from that submission.

Round 1

Reviewer 1 Report

Thank you for letting me review your revised manuscript. Unfortunately the authors only revised minor comments from the previous review round:

”Thank you for letting me review this interesting manuscript. The authors present a perspective on establishing a multidisciplinary liver tumor board (MDLTB) in a multi-site setting. However, apart from the thoroughly relevant subject matter, the manuscript has some weaknesses.

General comments:

  1. The manuscript available for review does not contain any figures, although they are mentioned in the main text. In addition to the complete lack of figures, the text also refers only to figure 3 and 4. (what about figure 1 and 2?)
  2. The provided checklist is confusing in its current form and should be restructured.
  3. The role of a clinical champion should be explained further.
  4. The implementation of the multi-site tumor board was not described appropriately (conference system, data protection etc.)
  5. Overall, the description of what is needed to establish a successful MDBLT remains superficial.

PbP Response:

  1. We have included the figures now as a separate pdf file, we apologize for any inconvenience this might have caused.

  2. We have now reconstructed the checklist, hopefully this will address the reviewer’s concerns.

  1. We believe clinic champion is someone who knows how to navigate the VA system and strongly believes in the program and is also invested in liver cancer and rallies disciplines to see it through and has a successor to hand it off.

  2. Multiple platforms have been used specially during COVID-19 pandemics. VA system is compatible with use of Microsoft teams and webex, although virtually any platform accessible and approved by the institution can be used. In case of VA system, teams have been approved as a platform to share data and it is considered secure. VA also offers intranet SharePoint sites with user level access control, but this can also be set up in any institution. For example, outside of VA, many health care systems use redcap with user level access control. We did not specify the platform to develop tumor board and share patient information as these would be unique to each center. Ultimately, these platforms should be approved by institutional IT.

  3. We have made the above clarifications as well as restructuring of our checklist table which hopefully can better help with understanding the process of MDTB establishment.

  4. This has been changed

  5. This has been changed

________

  1. Unfortunately I cannot see any figures. The provided documents do not include any figures and there is no additional data available for review. However, even if figures were included. There are only references for figure 3 and figure 4 - What about figure 1 and 2?
  2. Unfortunately again, I do not see any revised checklist.
  3. This should be explained in the manuscript.
  4. This should also be explained in the manuscript.
  5. Without any major revisions, the description of what is needed to establish a successful MDBLT still remains superficial.

Correction/Additional Review after reviewing the figures: (27.06.2021)

Thanks to the editorial board for sending me the figures.

Figure 3 illustrates the disciplines required to participate in an MDLTB. However, in my opinion the circular flow chart is not the best choice. The “ancillary staff” section has also to be rearranged to improve readability. 

Figure 4 nicely illustrates the distribution of liver cancer tumor boards across the US. However, in section B (History of liver cancer care at the VA), paragraph 2, the authors describe 18 regional VISN. Figure 4 reports 23 VISN. This discrepancy should be corrected.

Figures 1 and 2 are not mentioned in the main body. In line 165, the authors refer to “figure 3” regarding survival, which obviously should be figure 2. Regarding the results of VISN2, it should also be mentioned, that there was no difference in 5-year survival rate (figure 2) as mentioned in reference 14 (abstract only).

In conclusion, including the figures into the review did not change my final decision to suggest rejecting this manuscript due to the previously mentioned concerns regarding the quality and the missing benefit for the readership of cancers. 

Reviewer 2 Report

  1. Figure 1 and 2 should like to the text in in the manuscript.  I still can`t find them in the manuscript file.  
  2. Please consider removing the bottom part of figure 2.
  3. I have few suggestions for figure 3. 

Some words start with upper and some with lower case. Please make it consistent.

Please consider adding bullet points for each type of ancillary staff.

Make it consistent between department name vs person (e.g. Diagnostic radiology rather than radiologist,  Pathology  rather than pathologist, Hepatobiliary/Transplant surgery rather than surgery team).